



# Evaluation of a Quasi-steady state approximation of the cloud Droplet Growth Equation (QDGE) scheme for aerosol activation in global models using multiple aircraft data over both continental and marine environments

Hengqi Wang[1], Yiran Peng[1], Knut von Salzen[2], Yan Yang[3], Wei Zhou[3], Delong Zhao[3]

[1]Ministry of Education Key Laboratory for Earth System Modeling, Department of Earth System Science, Tsinghua University, Beijing, 100084, China
[2]Canadian Centre for Climate Modelling and Analysis, Environment and Climate Change Canada, Victoria, British Columbia, Canada
[3]Beijing Weather Modification Office, Beijing, 100101, China

*Correspondence to*: Yiran Peng (pyiran@mail.tsinghua.edu.cn) and Knut von Salzen (Knut.vonSalzen@canada.ca)

**Abstract.** This research introduces a numerically efficient aerosol activation scheme and evaluates it by using stratus and stratocumulus cloud data sampled during multiple aircraft campaigns in Canada, Chile, Brazil, and China. The scheme employs a Quasi-steady state approximation of the cloud Droplet Growth Equation (QDGE) to efficiently simulate aerosol

activation, the vertical profile of supersaturation, and the activated cloud droplet number concentration ($CDNC$) near the cloud base. We evaluate the QDGE scheme by specifying observed environmental thermodynamic variables and aerosol information from 31 cloud cases as input and comparing the simulated $CDNC$ with cloud observations. The average of mean relative error ($\overline{MRE}$) of the simulated $CDNC$ for cloud cases in each campaign ranges from 17.30 % in Brazil to 25.90 % in China, indicating that the QDGE scheme successfully reproduces observed variations in $CDNC$ over a wide range of

different meteorological conditions and aerosol regimes. Additionally, we carried out an error analysis by calculating the Maximum Information Coefficient (MIC) between the mean relative error ($MRE$) and input variables for the individual campaigns and all cloud cases. MIC values are then sorted by aerosol properties, pollution level, environmental humidity, and dynamic condition according to their relative importance to $MRE$. Based on the error analysis we found that the magnitude of $MRE$ is more relevant to the specification of input aerosol pollution level in marine regions and aerosol

hygroscopicity in continental regions than to other variables in the simulation.

## 1 Introduction

Aerosols play an important role in affecting the radiation balance of the earth-atmosphere system by scattering and absorbing shortwave radiation and altering the cloud reflectivity and lifetime (Twomey, 1974, 1977; Ghan, 2013; Forster et al., 2016;





Ramaswamy et al., 2019; Wang et al., 2020). Aerosol-cloud interactions remain as one of the largest sources of climate

modeling uncertainty (Intergovernmental Panel on Climate Change, 2013).

Aerosol-cloud interactions are largely driven by the activation of aerosols to form cloud droplets. The addition of activated aerosol to existing clouds can directly change the concentration and size of cloud droplets and thereby affect the microphysical properties and radiative forcing of the clouds. Aerosol activation is controlled by rapid and nonlinear aerosol and cloud microphysical processes (Meskhidze et al., 2005), which have not been explicitly resolved in climate models yet

(Fountoukis et al., 2007; Kang et al., 2015). Nenes et al. (2001) pointed out that the cloud droplet activation process is subject to kinetic limitations, including inertial, evaporation, and deactivation mechanisms, which further adds to the complexity of the aerosol activation.

Early parameterizations of aerosol activation in climate models were based on observations and derived through parameter fitting, using the aerosol number or mass concentration or other Cloud Condensation Nuclei (CCN) proxies (e.g., sulfate

mass) to empirically determine the activated $CDNC$ (Jones et al., 1994; Boucher and Lohmann, 1995; Jones and Slingo, 1996; Lohmann, 1997; Kiehl et al., 2000; Menon et al., 2002). Although these parameterizations have the advantages of convenience and low computational burden (Fountoukis et al., 2007), substantial uncertainties are resulting from limited spatiotemporal representativeness and unresolved variations in aerosol properties (Meskhidze et al., 2005). In the recent two decades, physically-based parameterization schemes of aerosol activation have emerged (Abdul-Razzak and Ghan, 2000;

Cohard et al., 2000; Fountoukis and Nenes, 2005; Ming et al., 2006; Kivekäs et al., 2008; Khvorostyanov and Curry, 2009; Shipway and Abel, 2010; Zhang et al., 2015). These schemes are based on the Köhler theory and are used in climate models to parameterize aerosol activation near the cloud base. As Köhler theory fundamentally describes the process by which water vapor condenses and forms liquid cloud droplets, it can be applied to a wide range of atmospheric conditions and aerosol pollution levels. However, considerable approximations of the Köhler theory are employed for application in climate models,

which leads to potential biases in comparison with results from more rigorous and accurate simulations of cloud droplet growth with adiabatic parcel models (e.g. Ghan et al. (2011)). The ongoing increase in computing power (Herrington and Reed, 2020) reduces the need to apply rigorous approximations of physicochemical processes in climate models. In the following, we will introduce a Quasi-steady state approximation of the cloud Droplet Growth Equation (QDGE) that provides an efficient alternative to parameterizations of activated $CDNC$ in climate models.

Parameterization schemes of aerosol activation were often evaluated with adiabatic parcel model simulations. These models explicitly solve aerosol activation and droplet growth processes by mimicking vertical uplifting of an air parcel containing a specified number of aerosol particles, predicting changes in temperature, humidity/supersaturation, activation of aerosols, and droplet growth from the cloud base upward. When utilizing identically specified aerosols, the results of a parcel model can be used as a benchmark to evaluate parameterizations. This approach has been extensively used to evaluate activation

schemes (Table 1). Alternatively, a less commonly used approach is to evaluate parameterizations by conducting a "closure experiment", that is, to carry out a parameterized calculation by specifying observed aerosol concentrations and





environmental thermodynamic conditions, and then compare the calculated and observed $CDNC$ (e.g. Snider and Brenguier, 2000; Guibert et al., 2003; Fountoukis and Nenes, 2005; Kivekäs et al., 2008). Though some parameterizations have been evaluated based on comparisons of simulated and observed $CDNC$ from aircraft campaigns, mostly regional data sets were
used for very specific meteorological conditions and pollution levels. It is essential to select a wide range of cloud data for different atmospheric conditions and pollution levels to arrive at meaningful conclusions for global climate model simulations.

In this study, we introduce the QDGE scheme and evaluate it by using cloud data from multiple aircraft campaigns in four different regions over the world, covering marine and continental conditions. This paper is organized as follows. The next
section describes the QDGE scheme and Sect. 3 summarizes the data and method used for the closure experiment and the evaluation. Section 4 illustrates the results of the closure experiment and analyzes the sources of simulation errors, followed by conclusions and discussion in Sect. 5.

**Table 1**. A summary of activation parameterizations and the evaluation methods in previous studies.

| Parameterization | Evaluation methods |
|---|---|
| Abdul-Razzak et al. (1998) | Parcel model |
| Cohard et al. (2000) | Parcel model |
| Snider et al. (2003) | Aircraft measurements |
| Fountoukis and Nenes (2005) | Parcel model; Aircraft measurements |
| Ming et al. (2006) | Parcel model |
| Kivekäs et al. (2008) | Other parameterizations; Aircraft measurements |
| Khvorostyanov and Curry (2009) | Twomey power law (Pruppacher et al., 1998) |
| Shipway and Abel (2010) | Parcel model |

## 2 QDGE scheme

Aerosol particles that are suspended in a parcel of air activate and grow into cloud droplets by condensation of water vapor if supersaturation with respect to water exceeds a critical value. In stratus and convective clouds, aerosol activation is particularly efficient in the vicinity of the cloud base, where supersaturation typically reaches its local maximum. Although observations provide evidence that aerosol activation is not limited to the region near the cloud base, this is omitted in the aerosol activation scheme described here, similar to most parcel models and parameterizations.

In order to determine the portion of the aerosols that activates and forms cloud droplets, a numerically efficient solution of the condensational droplet growth equation (e.g. Seinfeld and Pandis, 2016) is employed to simulate the growth of an ensemble of aerosol particles near the cloud base. The water vapor saturation ratio and cloud droplets above the cloud base are simulated by assuming a parcel of air with aerosols from below the cloud base, which ascends vertically to produce supersaturated conditions above the cloud base. The vertical velocity of the parcel of air, $w_c$ (in m $s^{-1}$), is either specified or
parameterized, as described in Sect. 3.2.3.





The change in wet aerosol particle radius, $R_{pw}$ (in m), by condensation of water vapor as a function of the water vapor saturation ratio ($S$, e.g. Emanuel, 1994) in the scheme is given by

$$R_{pw}\frac{dR_{pw}}{dt} = \frac{S-S_p}{C}, \tag{1}$$

where $S_p$ is the water vapor saturation ratio directly over the surface of the particle, which is obtained from $\kappa$-Köhler theory
(Petters and Kreidenweis, 2007),

$$S_p - 1 = \frac{A}{R_{pw}} - \frac{B}{R_{pw}^3}, \tag{2}$$

with the following parameters, which account for thermodynamic conditions in the cloud and physiochemical properties of the aerosol particles and droplets,

$$A = \frac{2M_w\sigma}{RT\rho_w}, \tag{3}$$

$$B = \kappa R_p^3, \tag{4}$$

and

$$C = \frac{\rho_w RT}{e_* D_v' M_w} + \frac{L_v\rho_w}{K_a' T}\left(\frac{L_v M_w}{RT} - 1\right), \tag{5}$$

where $\kappa$ is the aerosol hygroscopicity, $\sigma$ the surface tension of the solution/air interface (which is approximated by the surface tension of water here), $\rho_w$ the density of water, $M_w$ the molecular weight of water, $R$ the universal gas constant, $T$ the temperature, $R_p$ the dry aerosol particle radius, $e_*$ the saturation vapor pressure, $L_v$ the latent heat of vaporization, $K_a'$ the modified thermal conductivity of air accounting for non-continuum effects, $D_v'$ the modified diffusivity of water vapor in air accounting for non-continuum effects (Seinfeld and Pandis, 2016). Petters and Kreidenweis (2007) and Kreidenweis et al. (2008) provided tabulated values of the hygroscopicity parameter $\kappa$ for a variety of chemical compounds, based on laboratory data and modeling. They found that parameterized water contents are often within experimental uncertainty. However, the accuracy of this approach tends to decrease with decreasing aerosol water content. In particular, simulations of highly concentrated, non-ideal aqueous solutions with strong electrostatic interactions between ions with the Aerosol Inorganic Model (AIM; Wexler and Clegg (2002); http://www.aim.env.uea.ac.uk/aim/aim.html) give evidence for systematically different results at low aerosol water contents for some compounds (Kreidenweis et al., 2008). In order to improve biases at low relative humidity, the original method was extended to account for variations in $\kappa$ with relative humidity in the QDGE scheme. Specifically, piecewise-linear relationships between $\kappa$ and aerosol water activity for different chemical components were determined based on results from AIM.

Direct numerical solutions of Eq. (1) are computationally expensive, given that the condensation rate of water vapor depends on the aerosol size distribution and chemical composition, which leads to the highly non-linear behavior of the water vapor saturation ratio vertical profile. Typically, time steps much shorter than 1 second is required to solve this equation, which implies computational expenses that would prohibit applications in climate models (Khain et al., 2015). However, numerical





efficiency can be achieved by using a Quasi-steady state approximated Droplet Growth Equation (QDGE), which can be derived by using the local approximation $S \approx \mathrm{const}$ in Eq. (1), which can be conveniently expressed as follows,

$$\frac{dx}{du} = \delta - a\left(\frac{b}{x^{1/2}} - \frac{1}{x^{3/2}}\right),\tag{6}$$

for the time period from $t$ to $t + \Delta t_s$, with variable substitutions for particle size, $x = R_{pw}^2/2$, and time, $u = t|S - 1|/C$,

and parameters

$$\delta = \begin{cases} -1 \;, & \text{if } S < 1\;, \\ 1 \;, & \text{if } S \geq 1\;, \end{cases}\tag{7}$$

$$a = \frac{B}{2^{3/2}|S-1|}\;,\tag{8}$$

$$b = \frac{2A}{B}\;.\tag{9}$$

In the QDGE aerosol activation scheme, numerical efficiency is achieved by using pre-calculated solutions $x(u)$ of Eq. (6),

which are provided in the form of look-up tables (LUTs), for different values of $a$ and $b$. The $S$-dependent parameters $a$ and $\delta$, and $u$, are determined through an iterative procedure, for each time step and vertical level near cloud base, as described in the following.

A vertical grid with $N_{sub}$ sub-levels and grid spacing $\Delta z_s = \Delta z/N_{sub}$ is employed in the QDGE scheme, where $\Delta z$ is the grid spacing in the atmospheric host model, near cloud base. Calculations are only performed for the first host model grid layer

above the cloud base, with typical values $\Delta z_s \approx 1 - 10$ m, to ensure that the supersaturation maximum is captured and sufficiently well resolved in model applications of the aerosol activation scheme.

The growth calculations are performed for a sub-ensemble of aerosol particles which are selected from the full dry aerosol size distribution at regular size intervals, $\Delta \chi = 1/p\Delta\varphi$, where $p$ is on the order of 5 - 20 and $\Delta\varphi$ is the simulated particle size range of Aitken and accumulation mode aerosols, expressed in terms of a dimensionless particle size parameter $\varphi = $

$\ln(R_p/R_0)$, with $R_0 = 10^{-6}$ m. In this study, we set $p$ to 6, meaning that 6 discrete aerosol particle sizes are used. Sizes of other particles in the continuous aerosol size distribution are obtained from linear interpolation between the sizes of the particles in the discrete 6-member sub-ensemble.

$S$ and the dependent parameters in Eq. (6) are obtained through an iterative calculation, which explicitly requires the conservation of mass and energy. Specifically, a value of $S$ is first specified ("best guess" estimate) and Eq. (6) is integrated

over the time step $\Delta t_s = \Delta z_s/w_c$ to obtain a first estimate of the particle sizes at $z + \Delta z_s$. Next, an integration over the particle mass size distribution yields a first estimate of the liquid water mixing ratio, $LWC$, at $z + \Delta z_s$, subject to the initially specified value of the water vapor saturation ratio. Secondly, the total water mass mixing ratio, $r_t$, and liquid water static energy, $h$, in the ascending parcel of air are calculated, as defined by,

$$r_t = r_v + LWC\;,\tag{10}$$

$$h = gz + c_p T - L_v LWC\;,\tag{11}$$





Here, $r_v$ is the water vapor mass mixing ratio, $g$ the gravitational constant, and $c_p$ the heat capacity at a constant pressure of dry air. Currently, only adiabatic processes are considered, and therefore total water and liquid water static energy in Eqs. (11) and (12) are conserved as the parcel ascends from $z$ to $z + \Delta z_s$, with initial values for dry conditions from below cloud base. Consequently, the first estimates of $r_v$ and $T$ are determined from Eqs. (10) and (11), using the first estimate of $LWC$,
as described above. Subsequently, these results are used to update the water vapor saturation ratio, based on the standard definition of the water vapor saturation ratio,

$$S = \frac{r_v}{r_*} \left( \frac{1 + \frac{r_*}{0.622}}{1 + \frac{r_v}{0.622}} \right), \tag{12}$$

where $r_*$ is the saturation water vapor mass mixing ratio in the parcel of air, which depends on $T$. Subsequently, the updated value and the initial estimate of $S$ are compared and are used to determine an improved estimate of $S$ using a bisectional
method that minimizes the difference between different available estimates of $S$ through iteration. The method quickly converges to a desired value of $S$, which solves Eq. (6) and satisfies all necessary constraints according to Eqs. (10), (11), and (12). After the iterations are complete and results are available at $z + \Delta z_s$, the calculations are repeated in order to obtain $S$ at the next higher level above until results are available at all $N_{sub}$ levels.

Finally, the maximum value of the simulated vertical water vapor saturation ratio profile, $S_{max}$, is selected and used to
diagnose the critical particle size, which separates activated from non-activated particles, i.e. by requiring that $S_{max} = S_p$. Particles with sizes that are equal to or greater than the critical size are assumed to be activated. Consequently, the cloud droplet number concentration is obtained by integrating the activated particle size distribution accordingly. Above cloud base, a uniform vertical profile of the cloud droplet number mixing ratio is assumed, in good agreement with observations and detailed simulations of clouds (Gerber et al., 2008; Slawinska et al., 2012; Jarecka et al., 2013). Also, you can set the
entrainment rate to consider the effect of entrainment on the vertical profile if necessary.

The QDGE aerosol activation scheme has been previously used to assess Arctic indirect radiative forcing (Arora et al., 2015) and to determine the sensitivity of Arctic clouds to changes in future surface seawater dimethyl sulfide concentrations (Mahmood et al., 2019).

# 3 Data and methods

## 3.1 Campaign description

The worldwide cloud data used for the evaluation were sampled from four aircraft campaigns. The locations and instrument information of the four campaigns are shown in Fig. 1 and Table 2. The Canada (CAN) campaign provided marine stratus cloud data observed during the Radiation, Aerosol and Cloud Experiment (RACE) in fall 1995 off the coast of Nova Scotia, Canada (Peng et al., 2002). The Chile (CL) campaign provided marine stratocumulus clouds data observed during the



VAMOS Ocean-Cloud-Atmosphere-Land Study Regional Experiment (VOCALS-REx), for near-climatological atmospheric conditions off northern Chile and southern Peru (Wood et al., 2011). The Brazil (AMA) campaign provided continental stratus clouds data observed in Manaus, Brazil during the Green Ocean Amazon (GoAmazon2014/5) Experiment (Martin et al., 2016). The China (CN) campaign provided polluted continental stratus clouds data sampled in Beijing, China by the Beijing Weather Modification Office (Liu et al., 2020). These worldwide datasets comprise continental (CN and AMA),

coastal (CAN), and marine (CL) meteorological conditions. Additionally, they cover different levels of human influence on clouds, with an observed range of the mean aerosol number concentration ($N_a$) within 100 m below the cloud base from 282 $cm^{-3}$ to 1350 $cm^{-3}$.

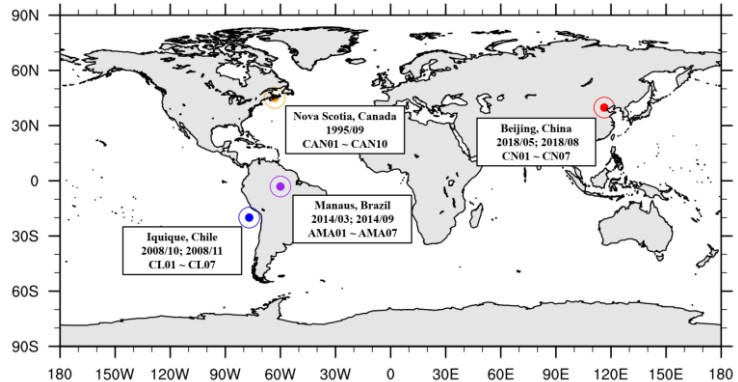

**Figure 1.** The geographical distribution of 31 selected cloud cases in the four aircraft campaigns. The text boxes provide the

locations, the periods, and the names of the cloud cases for each campaign.

**Table 2.** An overview of the four aircraft campaigns in this study.

| Name | CAN | CL | AMA | CN |
|---|---|---|---|---|
| Date | 1995/09 | 2008/10; 2008/11 | 2014/03; 2014/09 | 2018/05; 2018/08 |
| Location | Nova Scotia, Canada | Iquique, Chile | Manaus, Brazil | Beijing, China |
| Cloud type | stratus | stratocumulus | stratus | stratus |
| Campaign name | RACE | VOCALS-REx | GoAmazon2014/5 | / |
| CDNC instrument | FSSP (15 bins, 2.0~47.0 μm) | CAS (20 bins, 0.6~56.3 μm) | FCDP (20 bins, 1.5~150.0 μm) | FCDP (20 bins, 1.5~150.0 μm) |
| Aerosol instrument | PCASP (15 bins, 0.13~3.00 μm) ASAP (13 bins, 0.183~2.37 μm) | PCASP (30 bins, 0.09~3.00 μm) | PCASP (30 bins, 0.09~3.45 μm) | PCASP (30 bins, 0.10~3.00 μm) |
| Chemistry instrument | AMS | AMS | AMS | / |
| LWC instrument | King hot-wire probe | King hot-wire probe | King hot-wire probe and Johnson-Williams probe | King hot-wire probe |
| Atmospheric condition instrument | AIMMS | AIMMS | AIMMS | AIMMS |
| Number of selected cloud cases | 10 | 7 | 7 | 7 |




| Number of cases for $w_c$ calculation | 2 | 3 | 5 | 4 |
|---|---|---|---|---|
| $N_a$ | 476±294 cm$^{-3}$ | 282 ± 116 cm$^{-3}$ | 846 ± 819 cm$^{-3}$ | 1350 ± 916 cm$^{-3}$ |

Note: $N_a$ is the integrated number of particles detected by aerosol instruments and averaged within 100 m below the cloud base. The definition of cloud base and selection of cloud cases refer to Sect. 3.2.1. Calculation of $w_c$ refers to Sect. 3.2.3.

Aerosol and cloud measuring instruments utilized in the four campaigns are briefly presented in Table 2. The observed variables mainly include $CDNC$, $LWC$, the aerosol number-size distribution, the chemical compositions of aerosol, and atmospheric condition parameters. For the measurement of $CDNC$, the forward scattering spectrometer probe (FSSP) was used in the CAN campaign. The cloud, aerosol, and precipitation spectrometer (CAS) was used in the CL campaign. The fast cloud droplet probe (FCDP) was used in the AMA and CN campaigns. Although FCDP, FSSP, or CAS can observe cloud droplets with a particle size up to 150 µm, we only integrated the number for droplets with a particle size of 2 to 30 µm to derive the $CDNC$. Because cloud droplets larger than 30 µm are subject to collision-coalescence, and droplets smaller than 2 µm may be deactivated by evaporation (Fountoukis and Nenes, 2005). For the measurements of the $LWC$, the King hot-wire probe was used in all campaigns, and the Johnson-Williams probe was also equipped as an alternative option in GoAmazon2014/5. In terms of the aerosol observation, all the four campaigns utilized an onboard passive cavity aerosol spectrometer probe (PCASP), and some flights during the CAN campaign used the atmospheric solids analysis probe (ASAP), providing aerosol number concentration in multiple size bins roughly from 0.1 to 3 µm. We integrated the number for particles within the detected size range to determine $N_a$. In the CAN, AMA, and CL campaigns, the mass concentrations of aerosol chemical species, including $NH_4^+$, $NO_3^-$, $SO_4^{2-}$, $Cl^-$, and organics ($org$), were measured using the aerodyne aerosol mass spectrometer (AMS). The CN campaign lacked data for aerosol chemical composition (see Sect. 3.2.2). For the CL campaign, five aircraft (i.e. Lockhead C-130, BAe-146, Gulfstream-1, Dornier-228, and Twin Otter) carried out observations (Wood et al., 2011). In order to ensure data integrity and consistency for aerosol number-size distribution and chemical composition measurements in the subsequent analysis, we only selected data from the Gulfstream-1 flights. The atmospheric condition parameters ($T$, pressure ($P$), relative humidity ($RH$), vertical velocity ($w$)) were mainly observed by the airborne integrated meteorological measurement system (AIMMS), in all campaigns. For the CL campaign, vertical velocity data were not available from the Gulfstream-1 flights, thus we used the observed $w$ data from the Twin Otter flights that occurred simultaneous with Gulfstream-1 flights. Some meteorological variables that are required by the QDGE scheme, particularly including $r_v$, $r_t$, and $h$, were not available from the aircraft observations. Therefore, we calculated these based on other variables (Sect. 3.2.4). Detailed descriptions of the aforementioned observational instruments and data quality control procedures can be obtained from the relevant publications for the different aircraft campaigns (Li et al., 1998; Peng et al., 2002; Wood et al., 2011; Kleinman et al., 2012; Martin et al., 2016, 2017; Wang et al., 2020).





### 3.2 Data processing for closure experiment

#### 3.2.1 Data extraction

The flow chart of data extraction and processing is shown in Fig. 2. In the first step, we conducted a screening of observational data to obtain suitable cloud cases fulfilling the following conditions (Step 1 in Fig. 2). First, we selected cloud cases with continuous $LWC$ profile with $T > 0$ ℃ and $LWC \geq 0.05$ g cm$^{-3}$ in each layer, identifying the height of the cloud base as $H_{low}$ (see Fig. A1). Second, we checked whether the $LWC$ near the cloud base approximately satisfies the wet adiabatic assumption, that is, nearly free from entrainment. As shown in Fig. A1, we plotted the observed $LWC$ and the adiabatic $LWC$ ($LWC_{ad}$) profiles, the later ones were calculated by assuming that $LWC$ increases linearly with the height above cloud base ($H_c$), i.e. $LWC_{ad} = C_w H_c$. $C_w$ is the adiabatic liquid water lapse rate, which is a function of temperature (Brenguier, 1991). For liquid clouds, the value of $C_w$ varies from $0.5 \times 10^{-3}$ to $3.0 \times 10^{-3}$ g m$^{-4}$ (Peng et al., 2002). For the cases shown in Fig. A1, $C_w$ ranges from $0.6 \times 10^{-3}$ to $2.8 \times 10^{-3}$ g m$^{-4}$. The mean of $C_w$ in each cloud case is shown in Table A1. Considering that the entrainment rate was set to $1.0 \times 10^{-3}$ m$^{-1}$ (weak entrainment, Barahona and Nenes, (2007)) when running the QDGE scheme in order to be close to the real atmosphere, we identify the nearly adiabatic part in the cloud case (i.e. data sampled between $H_{low}$ and $H_{high}$ in Fig. A1) for obtaining the observed cloud properties for evaluating the simulation. Third, we excluded the impact of collision-coalescence in the selected cloud cases, by ensuring that the water contents of cloud droplets with size greater than 30 μm were less than $0.05$ g cm$^{-3}$. Finally, we checked to make sure each cloud case has $N_a$ larger than $CDNC$. Ultimately, we obtained 31 eligible cloud cases, as shown in Fig. A1. Table A1 listed the observed data in the selected cloud cases, $CDNC_O$ and $LWC$ were averaged over the adiabatic part of each cloud case, $N_a$ and $RH$ were averaged within 100 m below the cloud base.

As shown in Step 2 of Fig. 2, we classified data samples of each cloud case into cloudy and clear conditions by utilizing the following criteria. Data sampled inside the cloud (cloudy condition) requires that $LWC \geq 0.05$ g cm$^{-3}$, $CDNC > 10$ cm$^{-3}$, and $RH \geq 99.5$ %, and data samples outside the cloud (clear condition) requires that $LWC < 0.05$ g cm$^{-3}$, $N_a > 10$ cm$^{-3}$, and $RH < 99.5$ %.

During each flight, the sampling along the horizontal flight track was continuous, which allowed us to better characterize the cloudy conditions or atmospheric conditions inside or outside the cloud. In all the 31 selected cloud cases, we were able to extract data samples at $nl$ levels ($l_i, i = 1, 2, \ldots, nl$ from the cloud base; where $nl$ is usually 4, at least 2.) along horizontal flight tracks in each cloud case, and calculated the mean value of the observed variable $v$ ($V_{v,l_i}$) along the horizontal track in each level $l_i$. $V_{v,l_i}$ is then extended to the vertical model levels ($L_j, j = 1, 2, \ldots, NL$; where $L_j$ refers to the interfaces of the vertical layers in the model, i.e. $\Delta z = L_{j+1} - L_j$) for running the QDGE scheme, which is Step 3 as shown in Fig. 2. The extension proceeded with the following rules: The meteorological variables profile in clear condition, such as $T$, $P$, and $r_t$, were extended downwards to the surface by using hydrostatic equation and ideal gas law, then extended to the top by linear





extrapolation, and interpolated between $l_1$ and $l_{nl}$. The aerosol mass and number profiles were extended to surface and top by linear extrapolation and interpolated between $l_1$ and $l_{nl}$. RH was filled between $l_1$ and $l_{nl}$ by linear interpolation.

For each cloud case, the data samples in the clear air were used to obtain aerosol-related input information for the model simulations (number and mass concentrations of aerosol components in different particle size sections) and the profiles of

250    meteorological parameters. The data samples in cloudy conditions were used to obtain the vertical velocity and $LWC$ as input for the model, and to provide measured $CDNC$ for comparisons with model results and closure verification. These are Steps 4, 5, and 6, as shown in Fig. 2 and described in the next three subsections.

**Figure 2.** A flow chart to schematically show the data extraction and processing for this work.



### 3.2.2 Aerosol data for input

In each of the cloud cases from the different aircraft campaigns, aerosol number concentrations $N_{a\_p}$ ($p = 1, \ldots, np$; where $np$ is the number of size bins detected in observation, see Table 2) sampled by ASAP or PCASP were categorized in 13, 15, or 30 bins. The size-resolved aerosol number concentrations were subsequently interpolated to a common particle size distribution (PSD) with 6 prescribed size sections for model input based on the following method (as depicted in Fig. 3). First, we used the aerosol number concentration in each size bin of the PCASP (or ASAP) data to fit a continuous PSD using cubic spline interpolation (Fig. 3b). Second, we integrated the fitted PSD to obtain the aerosol number concentration $N_{a\_k}$ ($k$=1, … , 6) in the aerosol size sections employed by the QDGE scheme (the dry aerosol particle radius boundaries are at 0.050, 0.088, 0.155, 0.274, 0.483, 0.851, 1.500 µm, as shown in Fig. 3c). By utilizing this method, the total $N_a$ obtained by integration over the 6 QDGE sections was slightly different from the observed total aerosol number due to the fitting of PSD, thus we further weighed the total fitted aerosol number concentration by the observed aerosol number to ensure the conservation of total number concentration (i.e., the total $N_a$ integrated over the QDGE sections in Fig. 3c is the same as the aerosol number integrated over the observed PSD in Fig. 3a). Finally, the PSD of the aerosol number concentration in 6 sections (Fig. 3c) was used as input to the QDGE scheme.

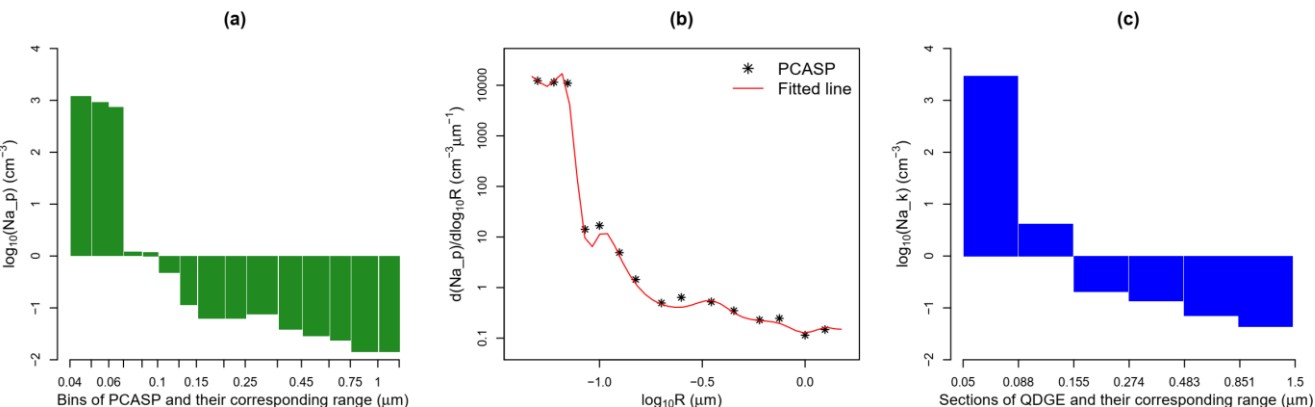

**Figure 3.** The processing of the observed aerosol number-size distribution for the input to the QDGE scheme. (a) shows the observed aerosol number concentration in each size bin sampled by PCASP, (b) the particle size distribution curve (red line) fitted to the observations (the asterisks refer to the observations that were derived from (a)), and (c) aerosol number concentration in 6 size sections, as prescribed in model simulations with the QDGE scheme.

For each of the CAN, AMA, and CL campaigns, the AMS provided measurements of chemical components over the entire campaign, providing concentrations of $NH_4^+$, $NO_3^-$, $SO_4^{2-}$, $Cl^-$, and $org$. The various chemical components in the aerosol were assumed to be internally mixed, thus different components share the same aerosol number concentration in each size section. To obtain the PSD of mass concentration of each chemical component, we made use of the AMS measurements. For



continental campaigns such as CN and AMA, we assumed that aerosols are composed of $NH_4NO_3$, $(NH_4)_2SO_4$, $NH_4Cl$, and organics ($Org$) (Shilling et al., 2018; Zhou et al., 2019; Li et al., 2020). For coastal or oceanic campaigns such as CAN and

CL, we took sea salt ($NaCl$) into account, too. For the CAN, AMA, and CL campaigns, we converted the AMS data of ion mass ($AMS_{ci}$, $ci$ is $NO_3^-$, $SO_4^{2-}$, $Cl^-$, or $org$)  to the mass of each chemical component ($m_c$, $c$ is $NH_4NO_3$, $(NH_4)_2SO_4$, $NH_4Cl$, organics ($Org$), or $NaCl$).

$$m_{NH_4NO_3} = \frac{AMS_{NO_3^-}}{M_{NO_3^-}} M_{NH_4NO_3}, \qquad (13)$$

$$m_{(NH_4)_2SO_4} = \frac{AMS_{SO_4^{2-}}}{M_{SO_4^{2-}}} M_{(NH_4)_2SO_4}, \qquad (14)$$

$$m_{NH_4Cl} = \frac{(1-\alpha)AMS_{Cl^-}}{M_{Cl^-}} M_{NH_4Cl}, \qquad (15)$$

$$m_{NaCl} = \frac{\alpha AMS_{Cl^-}}{M_{Cl^-}} M_{NaCl}, \qquad (16)$$

$$m_{Org} = AMS_{org}, \qquad (17)$$

where $M_{ci}$ and $M_c$ are the molecular weight of ion $ci$ and chemical component $c$, respectively. Here we assume that concentrations of $NH_4^+$ are sufficiently high to balance all anions. The mass of sea salt in different campaigns is controlled

by a given factor $\alpha$ to partition the amount of $Cl^-$ in sea salt and continental chemical components. We set the values of $\alpha$ as 0, 90%, and 95% for AMA, CAN, and CL campaigns. That is, 90% and 95% of $Cl^-$ are attributed to sea salt in the coastal campaign CAN and the oceanic campaign CL, respectively. Based on the calculated mass concentration of each chemical component, the average density of aerosol can be obtained:

$$\rho_a = \frac{\sum_{c=1}^5 m_c}{\sum_{c=1}^5 m_c/\rho_c}, \qquad (18)$$

where $\rho_c$ is the density of each component $c$, and they are 1725, 1769, 1527, 1900, and 1400 kg m$^{-3}$ for $NH_4NO_3$, $(NH_4)_2SO_4$, $NH_4Cl$, $NaCl$, and $Org$, respectively (Ferek et al., 1998; Nakao et al., 2013). Consequently, we can obtain the mass concentration (unit kg cm$^{-3}$) of each component $c$ in section $k$ following this equation:

$$Mass_{c,k} = \frac{m_c}{\sum_{c=1}^5 m_c} \cdot N_{a\_k} \frac{4\pi}{3} R_k^3 \rho_a, \qquad (19)$$

where $R_k$ is the median radius of section $k$.

Since no AMS data are available for the CN campaign, we assumed the mass fraction of different chemical components according to contemporaneous measurements in Beijing, China (Zhou et al., 2019; Li et al., 2020), as shown in Table A2. Under the assumption of $\rho_a = 1600$ kg m$^{-1}$ (Levy Zamora et al., 2019), $Mass_{c,k}$ in the CN campaign can be obtained from Eq. (19).

Finally, we obtained the number concentration of total aerosol and the mass concentration of each chemical component from

PCASP/ASAP and AMS measurements in each cloud case and calculated aerosol number and mass concentrations in 6





prescribed size sections following the above procedures (Step 4 in Fig. 2). We then used the aerosol information as input to drive the QDGE scheme.

### 3.2.3 Vertical velocity for input

The averaged updraft velocity ($w_+$) and sub-grid vertical velocity ($w_{sub}$) obtained from the observed vertical velocity ($w$)

samples in clouds were used to calculate $w_c$ ($w_c = w_+ + w_{sub}$) as input for running the QDGE scheme (Step 5 in Fig. 2). The updraft velocity is a key variable for parameterizing aerosol activation. Peng et al. (2005) pointed out that using a characteristic value of the vertical velocity distribution (0.8 times the standard deviation of the distribution) is a good approximation for simulating the nucleated cloud droplet number of marine stratus when running the parcel model. Meskhidze et al. (2005) also gave a method to calculate $w_+$, which had the optimal closure for cumulus and stratocumulus

clouds. Here, we derived a universal method for calculating $w_+$ in stratus and stratocumulus based on the above two studies.

According to Meskhidze et al. (2005), the averaged updraft velocity ($w_+$) can be calculated by probability density function (PDF) of $w$, $p(w)$:

$$w_+ = \frac{\int_0^\infty w p(w) dw}{\int_0^\infty p(w) dw} . \tag{20}$$

For the normal PDF with the mean velocity $w_0$ and standard deviation $\sigma$, $p(w)$ can be represented as

$$p(w) = \frac{1}{\sqrt{2\pi}\sigma} \exp\left(-\frac{(w-w_0)^2}{2\sigma^2}\right) = \beta\phi(\omega) , \tag{21}$$

where $\omega = \beta w + \gamma$, $\beta = 1/\sigma$, $\gamma = -w_0/\sigma$, and $\phi(\omega)$ is the standard normal PDF.

Take Eq. (21) into Eq. (20) and obtain

$$w_+ = \frac{\phi(\gamma)}{(1-\Phi(\gamma))\beta} - \frac{\gamma}{\beta} = \frac{\phi(\gamma)}{(1-\Phi(\gamma))}\sigma + w_0 , \tag{22}$$

where $\Phi(\gamma)$ is the cumulative distribution function of the standard normal PDF that can be represented by error function

(erf):

$$\Phi(\gamma) = \int_{-\infty}^{\gamma} \phi(t)dt = \frac{1}{2}\left(1 + \text{erf}(\frac{\gamma}{\sqrt{2}})\right). \tag{23}$$

Especially, when $w_0 = 0$,

$$w_+ = \frac{\phi(0)}{(1-\Phi(0))}\sigma = \sqrt{\frac{2}{\pi}}\sigma \cong 0.8\sigma, \tag{24}$$

which is consistent with the characteristic velocity pointed by Peng et al. (2005) used for assessing cloud droplet closure for

stratocumulus clouds sampled in the CAN campaign.

A sub-grid vertical velocity ($w_{sub}$) is needed for the QDGE scheme, and it can be derived from the square root of the Turbulent Kinetic Energy ($TKE$) following Morrison and Pinto (2005):

$$w_{sub} = \sqrt{\frac{2}{3}TKE}, \tag{25}$$





where the $TKE$ can be calculated according to its definition, which is half the sum of the variances (square of standard

deviations) of the velocity components:

$$TKE = \frac{1}{2}(\overline{(u')^2} + \overline{(v')^2} + \overline{(w')^2}),$$   (26)

In this study, we assume that no horizontal movement occurs in cloud during the horizontal flight tracks, that is, $\overline{(u')^2} = \overline{(v')^2} = 0$ and $\overline{(w')^2} = \sigma^2$. Therefore, the sub-grid vertical velocity can be represented by σ:

$$w_{sub} = \frac{\sigma}{\sqrt{3}}.$$   (27)

If the observed $w$ in each selected cloud case obeyed the normal distribution, we could calculate $w_c$ ($w_c = w_+ + w_{sub}$) following Eqs. (22) and (27) as input for running the QDGE scheme easily. We checked the normality of $w$ distribution by drawing a quantile-quantile (Q-Q) plot using the observed $w$ values along the horizontal flight track of the cloud case, taking CN01 as an example in Fig. 4. The linearity between the Q-Q plot of observed $w$ samples and a standard normal distribution indicates that $w$ data does indeed follow the normal distribution.

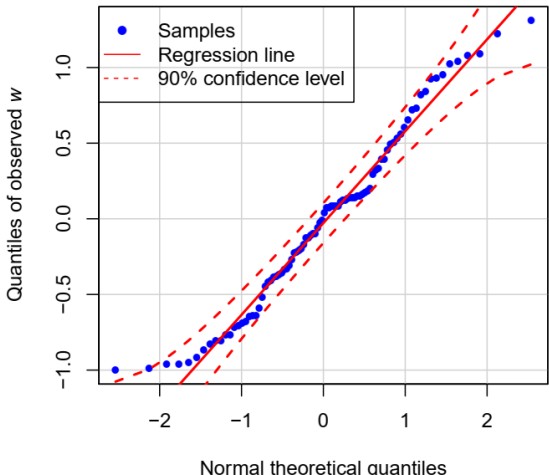


**Figure 4.** A normal quantile-quantile plot for comparing the observed $w$ sampled by aircraft in cloud case CN01 with a standard normal distribution. The linearity of the data points (blue) suggests that the observed $w$ are normally distributed.

In the four campaigns of this study, 4 cloud cases in CN, 2 cases in CAN, 5 cases in AMA, and 3 cases in CL have enough data samples to obtain the PDF of $w$ (Table 2), as plotted for checking the normality of $w$ distribution in Fig. A2. However,

the $w$ PDF in two of the CAN cloud cases does not conform to the normal distribution very well (panel (5) and (6) of Fig. A2). So, we used the mean and standard deviation of $w$ distribution in Peng et al. (2005) to obtain $w_c$ in the CAN campaign. For the CN, AMA, and CL campaigns, we directly calculated the $w_c$ from available data samples for the cloud cases plotted in Fig. A2 and used their mean values for cloud cases lacking enough $w$ values in each campaign (Table A1).





### 3.2.4 Meteorological input

Some meteorological variables ($T$, $P$, $RH$, and $LWC$) can be obtained from AIMMS measurements directly, though, others ($r_v$, $r_t$, and $h$) need to be calculated according to available variables (Step 6 in Fig. 2). We obtained $r_v$ by the following equation:

$$r_v = \frac{0.622 e_* RH}{P - e_*},\tag{28}$$

where $e_*$ can be estimated by referring to Murray (1967):

$$e_* = 6.1078 e^{\left(\frac{17.2694(T-273.16)}{T-35.86}\right)}.\tag{29}$$

Then, $r_t$ and $h$ can be obtained by Eqs. (10) and (11) from $r_v$ and other available variables. All meteorological variables were extracted and interpolated to model levels, as described in Sect. 3.2.1. The profiles of measured meteorological variables served as the initial state to drive the QDGE scheme.

### 3.2.5 Determination of $N_{sub}$

As mentioned in Sect. 2, the QDGE scheme simulates vertical profiles of supersaturation to determine $S_{max}$, for a vertical grid with the size $\Delta z_s = \Delta z/N_{sub}$, where $\Delta z$ is the grid size of the atmospheric host model. The accuracy of the simulated supersaturation profile generally increases with $N_{sub}$, though, large values of $N_{sub}$ imply higher computational burdens. For applications of the QDGE scheme in atmospheric models, it is therefore important to determine an optimal value of $N_{sub}$ that yields sufficiently accurate supersaturation profiles at acceptable costs.

Figure 5a plots the vertical profiles of $S$ simulated by the QDGE scheme with different $N_{sub}$ values for the cloud case CN01. The results show that each profile with $N_{sub} \geq 3$ produces a well-defined maximum of $S$ ($S_{max}$), which approaches to a stable value as $N_{sub}$ is further increased. All cases seem to converge to a similar value as $S_{max}$ with $N_{sub} = 150$, as plotted in Fig. 5a. Figure 5b shows the variation of $S_{max}$ with the increasing $N_{sub}$ for all cloud cases in the four campaigns. Overall, $S_{max}$ fluctuates dramatically with $N_{sub} < 10$, but plateaus when $N_{sub}$ is greater than 60 (10 for CAN). Results obtained for 375 $N_{sub} = 150$ and $N_{sub} = 60$ are similar. The mean relative error and correlation coefficient between $S_{max}$ with $N_{sub} = 150$ and that with $N_{sub} = 60$ are 1.97% and 0.9997, respectively. Therefore, we used $N_{sub} = 60$ in this study ($N_{sub} = 10$ for CAN). Further discussion regarding the selection of $N_{sub}$ are provided in Sect. 5.



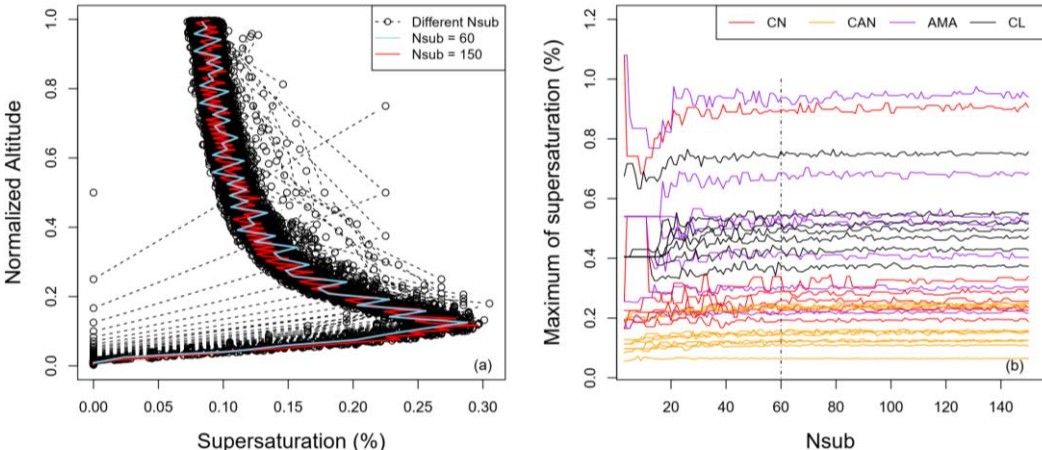

**Figure 5.** (a) Vertical profiles of the simulated supersaturation for different $N_{sub}$ (1-150) in the QDGE scheme for the cloud
case CN01. (b) Changes of the maximum supersaturation with different $N_{sub}$ for all cloud cases in the four campaigns.

### 3.3 Statistical parameters for evaluation and error analysis

The QDGE scheme simulates the $CDNC$ ($CDNC_M$) in each cloud case, based on $S_{max}$. Considering that aerosol activation is
particularly efficient in the vicinity of the cloud base in stratus and convective clouds, the QDGE scheme only calculates the
$CDNC$ at the cloud base (Sect. 2). Here, we considered the effect of weak entrainment on the vertical profile of the cloud
droplet number mixing ratio in order to be close to the real cloud base in the atmosphere (Sect. 3.2.1). Therefore, we
evaluated the simulation effect of the QDGE scheme by comparing $CDNC_M$ with the vertically average value of the observed
$CDNC$ ($CDNC_O$) in the nearly adiabatic part of the cloud (between $H_{low}$ and $H_{high}$ in Fig. A1) (Sect. 3.2.1). Correspondingly,
the mean bias ($MB$) and mean relative error ($MRE$) of each cloud case can be calculated, as follows:

$$MRE = |MB| = \left| \frac{CDNC_M - CDNC_O}{CDNC_O} \cdot 100\% \right|, \tag{30}$$

where $MRE$ of each cloud case will also be used for subsequent error analysis.

To evaluate the overall accuracy of the QDGE scheme, we also calculated the mean values of $CDNC_O$, $CDNC_M$, $MB$, $MRE$
for cloud cases in each campaign, namely $\overline{CDNC_O}$, $\overline{CDNC_M}$, $\overline{MB}$, and $\overline{MRE}$. Besides, the R square ($R^2$) between the $CDNC_O$
and $CDNC_M$ in each campaign was also calculated.

To quantify the contributions of different physical variables to errors in the simulated $CDNC$ with the QDGE scheme, we
calculated the Maximum Information Coefficient (MIC) (Reshef et al., 2011), which provides a measure for the strength of
the relationship between each input variable and $MRE$. MIC can be a good measure to capture the association between the
attributive variable and MRE for different types of relationships, such as linear, exponential and many complex functional





relationships (Reshef et al., 2011). There is no need to standardize the data before the MIC calculation and the calculations have low computational complexity and high robustness. However, it should be noted that the association here does not refer
to a specific correlation, such as temporal or spatial correlation, or positive or negative correlation, but refers to the strength of a certain relationship between the variable and MRE. The MIC value is always between 0 and 1. The higher the MIC value, the stronger the association between the input variable and $MRE$, that is, the input variable contributes more significantly to the $MRE$. Here, we calculated the MIC base on the minepy package in Python (Albanese et al., 2018), and set the parameters required in MIC as the default settings suggested by the code developers. Different parameters had an
insignificant effect on the relative importance of variables and MRE.

We calculated the MIC between $MRE$ and each one of the following input variables: the relative humidity ($RH$), the mean vertical velocity ($w_+$) and the sub-grid vertical velocity ($w_{sub}$) to represent environmental and dynamic conditions; the total aerosol number ($N_a$) as a proxy of pollution level; the hygroscopicity of aerosol ($K_m$) weighted by composition volume fraction, and the effective radius of aerosol PSD ($R_{e,a}$) to represent the chemical and size properties of the aerosol. Here, $K_m$,
and $R_{e,a}$ are defined as:

$$K_m = \frac{\sum_{c=1}^{5}\frac{m_c}{\rho_c}\kappa_c}{\sum_{c=1}^{5}\frac{m_c}{\rho_c}}, \tag{31}$$

$$R_{e,a} = \frac{\sum_{p=1}^{np} R_p^3 N_{a\_p}}{\sum_{p=1}^{np} R_p^2 N_{a\_p}}, \tag{32}$$

where $\kappa_c$, the hygroscopicity of component $c$, is accounted for variations with relative humidity in the QDGE scheme (Sect. 2). $R_p$ represents the middle radius in the $p^{th}$ particle size bin observed by PCASP or ASAP (see Sect. 3.2.2 and Table 2).
For MIC calculation, the values of input variables derived from observations are listed in Table A1 for each cloud case.

## 4 Results

### 4.1 Closure experiment

The results of the closure experiment are shown in Fig. 6. Almost all $CDNC_M$ values fall within 30 % of the mean observations in the clouds. $R^2$ is above 0.94 for all campaigns, which indicates a good agreement between simulation and
observation. For the four campaigns covering marine to continental conditions, the $\overline{MRE}$ values are all below 26 % and the $\overline{MB}$ values are within $\pm 20$ %. The AMA campaign produces the best agreement between model results and observations, with a $\overline{MRE}$ value of 17.30 %. On the other hand, the CN campaign produces a poor agreement, with a $\overline{MRE}$ value of 25.90 %. However, cloud droplet number concentrations are underestimated for all cloud cases for the CL campaign ($\overline{MB} = \overline{MRE} = -19.36$ %), which may be related to the high activation ratio ($AR$, the ratio of $N_a$ to $CDNC_O$, see Table A1) in this
region. $AR$ in all CL cases are higher than 60 %, suggesting that the marine environment is favorable for more aerosol



particles to be activated. If particles with a smaller size than the detection limit of PCASP (about 10 nm) are activated, it could lead to an underestimation of the simulated $CDNC$ in the CL campaign.

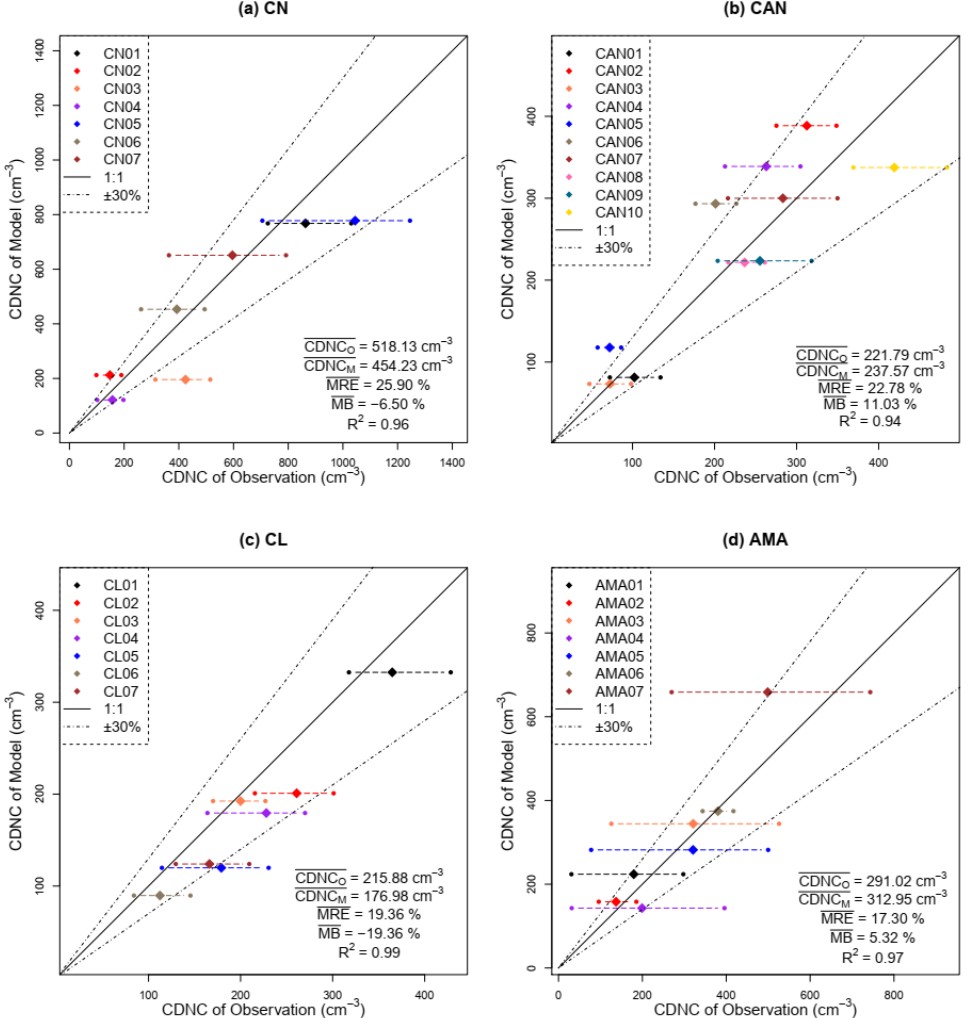

**Figure 6.** A closure experiment between $CDNC_O$ and $CDNC_M$ for each cloud case in the (a) CN, (b) CAN, (c) CL, and (d)
AMA campaigns. The horizontal dash lines represent the range of the observed $CDNC$ within the 25% and 75% quantiles.

In order to provide further context, we compare the $\overline{MRE}$ values of this study to previous studies with different aerosol activation parameterizations and aircraft measurements, as shown in Table 3. The $\overline{MRE}$ values are relatively high for those early parameterizations, basically around 50 %. In the recent two decades, the performance of physically-based parameterization has been significantly improved, as is evident from a reduction of the $\overline{MRE}$ to about 30 %. For instance, 435 one of the schemes (Fountoukis and Nenes, 2005) achieved remarkable closure (with $\overline{MRE}$ of 13.5 %) for continental



cumuliform/stratus. In this study, the QDGE scheme performs decently (the $\overline{MRE}$ values are all below 26 %) in four different regions, indicating that the scheme is suitable for simulations of cloud droplet number concentrations over a wide range of different meteorological conditions and different levels of aerosol pollution.

**Table 3.** Comparison of results from simulations with activation schemes and the QDGE method (Mainly referring to Fountoukis et al. (2007))

| Parameterization or Model | $\overline{MRE}$ (%) | Observed cloud type | Location | Reference |
|---|---|---|---|---|
| Flossmann et al. (1985) | ~50.00 | Continental stratocumulus | North of England | (Hallberg et al., 1997) |
| UWyo parcel model[a] | <50.00 | Marine stratocumulus | Tenerife, Spain | (Snider et al., 2003; Snider and Brenguier, 2000) |
| Fountoukis and Nenes (2005); Nenes and Seinfeld (2003) | ~30.00 | Coastal stratus | Monterey, California, USA | (Meskhidze et al., 2005) |
| Fountoukis and Nenes (2005) | 13.50 | Continental cumuliform /stratus | Cleveland and Detroit, USA | (Fountoukis et al., 2007) |
| Kivekäs et al. (2008) | ~35.00 | Continental stratus | North of Finland | (Kivekäs et al., 2008) |
| QDGE scheme | 17.30 | Continental stratus | Manaus, Brazil | This work |
| | 19.36 | Marine stratocumulus | Iquique, Chile | |
| | 22.78 | Costal stratus | Nova Scotia, Canada | |
| | 25.90 | Continental stratus | Beijing, China | |

a. UWyo parcel model, available at http://www.das.uwyo.edu/ccp/ web

## 4.2 Error analysis

Although the performance of the QDGE scheme is good in different aircraft campaigns, it is useful to analyze sources of biases in the simulations. Following the procedures described in Sect. 3.3, we calculated the Maximum Information Coefficient (MIC) between MRE and the input variables of the QDGE scheme, including aerosol properties ($K_m$, and $R_{e,a}$), thermodynamic state ($RH$), pollution level ($N_a$), and atmosphere dynamic conditions ($w_+$ and $w_{sub}$), as shown in Table A1. The MIC values for all cloud cases and each campaign have been shown in Table 4.

For almost all campaigns, the aerosol number concentration and the hygroscopicity, have the most significant impacts on $MRE$. This is consistent with the droplet growth equation, according to which the variation of supersaturation $S$ with height is essentially determined by the competition between the production of $S$ by adiabatic cooling and the reduction in $S$ from condensational growth of the particles, the latter mainly depends on the number and solubility of the aerosol particles. In detail, $N_a$ has a greater impact on $MRE$ in marine regions (CAN and CL), but $K_m$ is more significant in continental regions (CN and AMA). In marine regions, where $N_a$ is relatively low (Table 2), a small fluctuation in $N_a$ can cause noticeable changes in the simulated $S_{max}$ and CDNC, which makes $MRE$ more sensitive to $N_a$. However, in continental areas, $N_a$ is relatively high, and the change in hygroscopicity becomes more important to $MRE$. The atmospheric humidity and the dry size of the aerosol particle also have non-negligible impacts on $MRE$. Both affect the hygroscopic growth of aerosol particles





and the reduction in $S$. Overall, the atmosphere dynamic conditions have the most insignificant impact on $MRE$, which may be attributed to the weak variation of them in stratus and stratocumulus clouds (Table A1).

The MIC values also help to explain the relatively poor simulation performance of some campaigns. The chemical properties
of the aerosol, which affect $K_m$, are very important for the simulation in the continental region, but the CN campaign lacks AMS data and we applied the same chemical composition for all cloud cases, based on earlier measurements in this region (Sect. 3.2.2). Given the importance of the chemical properties, simultaneous measurements of chemical components probably would have helped to enhance the accuracy of simulated $CDNC$ for the CN campaign. Another possible cause of biases in simulated $CDNC$ for the CN campaign is a much larger standard deviation of observed $N_a$ (see Table 2) than that of
other campaigns, which could be responsible for the error in the simulated $CDNC$. However, it should be noted that although the CAN campaign is characterized by the presence of coastal clouds and smaller variations in $N_a$, its $MRE$ is higher than the AMA campaign, which may be related to the application of uniform updraft velocity in simulations for the CAN campaign (Sect. 3.2.3 and Table A1).

Overall speaking, the errors in the simulated CDNC is largely relevant to the missing data in observation (such as CN and
CAM campaign), the analysis of MIC and error sources here could provide a good reason to develop and improve measurement strategies in the future aircraft campaigns.

**Table 4.** The calculated MIC values between $MRE$ and different input variables for all cloud cases and each campaign.

| CN | | CAN | | CL | | AMA | | ALL | |
|---|---|---|---|---|---|---|---|---|---|
| $K_m$ | 0.522 | $N_a$ | 0.610 | $RH$ | 0.522 | $K_m$ | 0.522 | $N_a$ | 0.343 |
| $RH$ | 0.522 | $K_m$ | 0.396 | $N_a$ | 0.470 | $N_a$ | 0.522 | $K_m$ | 0.315 |
| $N_a$ | 0.470 | $R_{e,a}$ | 0.396 | $K_m$ | 0.292 | $w_+$ | 0.470 | $RH$ | 0.242 |
| $w_+$ | 0.470 | $RH$ | 0.396 | $R_{e,a}$ | 0.198 | $w_{sub}$ | 0.470 | $R_{e,a}$ | 0.202 |
| $w_{sub}$ | 0.470 | $w_+$ | 0.000 | $w_+$ | 0.198 | $RH$ | 0.292 | $w_+$ | 0.170 |
| $R_{e,a}$ | 0.292 | $w_{sub}$ | 0.000 | $w_{sub}$ | 0.198 | $R_{e,a}$ | 0.198 | $w_{sub}$ | 0.170 |

## 5 Conclusions and discussion

In this paper, we introduce a numerically efficient aerosol activation scheme, which calculates the maximum cloud
supersaturation and cloud droplet number concentration ($CDNC$) by employing a Quasi-steady state approximation of the cloud Droplet Growth Equation (QDGE) scheme. The QDGE scheme utilizes look-up tables and an iterative method for solving mass and energy budgets for efficient applications of the scheme in climate models. We evaluated the simulated $CDNC$ with worldwide cloud data sampled during four aircraft campaigns, covering a wide range of different meteorological conditions and different levels of aerosol pollution. The aerosol information, updraft velocity, and meteorological conditions
were carefully extracted from aircraft measurements and applied to drive the QDGE scheme. The simulated CDNC is compared with the observed correspondence in the nearly adiabatic part of the cloud, for evaluating the performance of the scheme. The average values of the mean relative error and the mean bias in the four campaigns are all within 26% and ±20%,





respectively, indicating that the QDGE scheme can reasonably simulate the activated $CDNC$ on a regional or global scale. We also investigated the potential sources of error in the simulated $CDNC$ and found that the magnitude of the mean relative

error is mostly relevant to the aerosol number concentration in marine regions and to aerosol hygroscopicity in continental regions than to other variables in the simulation.

Several points are worthy of mentioning for future work. The QDGE scheme can be further optimized in several aspects. First, $N_{sub} = 60$ generates reasonably good results in four different regions in this study, but this number is a little high and the computation will be too demanding to apply in general circulation models. Second, the iterative calculation to derive

supersaturation in each sub-grid level can be computationally expensive. Therefore, both adjustments on $N_{sub}$ number and optimization on the iteration would be necessary before the QDGE scheme is applied in the climate model. Last, we also want to evaluate the QDGE scheme by comparing it with parcel model simulations, to further identify the sources of error related to the approximations in the scheme. These works would be considered in future studies.

**Appendix A**

**Table A1.** A summary of observed ($CDNC_O$, $N_a$, $RH$, and $LWC$), derived ($AR$, $Sol$, $C_w$, $K_m$, $R_{e,a}$, $w_+$, and $w_{sub}$), simulated and evaluative ($CDNC_M$, $MB$, and $MRE$) variables of each cloud case in four campaigns.

| Case | Observed variables | | | | | | Derived variables | | | | | | Simulated and evaluative variables | | |
|---|---|---|---|---|---|---|---|---|---|---|---|---|---|---|---|
| | $CDNC_O$ (cm$^{-3}$) | $N_a$ (cm$^{-3}$) | $RH$ (%) | $LWC$ (g cm$^{-3}$) | $AR$ (%) | $Sol$ (%) | $C_w \times 10^{-3}$ (g cm$^{-4}$) | $K_m$ | $R_{e,a}$ (μm) | $w_+$ (m s$^{-1}$) | $w_{sub}$ (m s$^{-1}$) | $CDNC_M$ (cm$^{-3}$) | $MB$ (%) | $MRE$ (%) |
| CN01 | 863.25 | 3016.27 | 67.92 | 0.20 | 28.62 | 65.00 | 0.69 | 0.37 | 0.23 | 0.469 | 0.340 | 767.86 | -11.05 | 11.05 |
| CN02 | 148.17 | 372.77 | 61.89 | 0.06 | 39.75 | 65.00 | 0.71 | 0.39 | 0.41 | 0.609 | 0.441 | 212.3 | 43.28 | 43.28 |
| CN03 | 424.41 | 432.05 | 61.89 | 0.08 | 98.23 | 65.00 | 1.04 | 0.39 | 0.15 | 0.609 | 0.441 | 195.84 | -53.86 | 53.86 |
| CN04 | 157.49 | 1738.09 | 57.71 | 0.12 | 9.06 | 65.00 | 0.81 | 0.4 | 0.98 | 0.609 | 0.441 | 121.33 | -22.96 | 22.96 |
| CN05 | 1044.72 | 1550.93 | 88.12 | 0.43 | 67.36 | 65.00 | 1.99 | 0.33 | 0.18 | 0.714 | 0.516 | 777.82 | -25.55 | 25.55 |
| CN06 | 392.89 | 850.10 | 72.42 | 0.22 | 46.22 | 65.00 | 1.93 | 0.35 | 0.56 | 0.444 | 0.314 | 453.34 | 15.39 | 15.39 |
| CN07 | 596.01 | 1486.6 | 66.79 | 0.11 | 40.09 | 65.00 | 2.36 | 0.37 | 0.22 | 0.609 | 0.441 | 651.10 | 9.24 | 9.24 |
| CAN01 | 102.28 | 108.26 | 95.27 | 0.12 | 94.48 | 62.50 | 1.03 | 0.54 | 0.84 | 0.299 | 0.215 | 81.26 | -20.55 | 20.55 |
| CAN02 | 312.43 | 461.86 | 82.95 | 0.23 | 67.65 | 73.95 | 1.37 | 0.76 | 0.17 | 0.299 | 0.215 | 388.57 | 24.37 | 24.37 |
| CAN03 | 72.69 | 110.60 | 97.07 | 0.28 | 65.72 | 79.40 | 2.40 | 0.68 | 0.3 | 0.299 | 0.215 | 73.31 | 0.85 | 0.85 |
| CAN04 | 263.02 | 547.91 | 86.3 | 0.22 | 48.00 | 73.95 | 1.50 | 0.71 | 0.67 | 0.299 | 0.215 | 338.82 | 28.82 | 28.82 |
| CAN05 | 72.12 | 176.43 | 84.6 | 0.11 | 40.88 | 62.50 | 1.15 | 0.65 | 0.28 | 0.299 | 0.215 | 117.77 | 63.30 | 63.30 |
| CAN06 | 201.15 | 441.24 | 90.82 | 0.19 | 45.59 | 73.95 | 1.67 | 0.66 | 0.85 | 0.299 | 0.215 | 293.30 | 45.81 | 45.81 |
| CAN07 | 283.26 | 673.60 | 84.23 | 0.18 | 42.05 | 73.95 | 1.67 | 0.74 | 0.18 | 0.299 | 0.215 | 299.97 | 5.90 | 5.90 |
| CAN08 | 236.61 | 561.35 | 79.83 | 0.25 | 42.15 | 73.95 | 1.82 | 0.79 | 0.22 | 0.299 | 0.215 | 221.63 | -6.33 | 6.33 |
| CAN09 | 255.29 | 1064.55 | 79.83 | 0.26 | 23.98 | 73.95 | 1.51 | 0.79 | 0.31 | 0.299 | 0.215 | 223.57 | -12.43 | 12.43 |
| CAN10 | 419.06 | 609.57 | 81.25 | 0.21 | 68.75 | 73.95 | 0.62 | 0.78 | 0.12 | 0.299 | 0.215 | 337.48 | -19.47 | 19.47 |
| CL01 | 364.78 | 493.78 | 54.36 | 0.15 | 73.88 | 72.25 | 2.54 | 0.6 | 0.13 | 0.618 | 0.447 | 332.53 | -8.84 | 8.84 |
| CL02 | 260.91 | 339.76 | 64.86 | 0.13 | 76.79 | 84.79 | 2.70 | 0.59 | 0.13 | 0.537 | 0.389 | 200.93 | -22.99 | 22.99 |
| CL03 | 199.93 | 309.33 | 41.98 | 0.18 | 64.63 | 80.27 | 1.86 | 0.74 | 0.14 | 0.618 | 0.447 | 192.45 | -3.74 | 3.74 |
| CL04 | 227.94 | 272.76 | 40.43 | 0.09 | 83.57 | 70.36 | 1.53 | 0.96 | 0.13 | 0.618 | 0.447 | 179.44 | -21.28 | 21.28 |





| | | | | | | | | | | | | | | |
|---|---|---|---|---|---|---|---|---|---|---|---|---|---|---|
| CL05 | 179.08 | 187.54 | 57.02 | 0.19 | 95.49 | 79.45 | 2.06 | 0.63 | 0.12 | 0.618 | 0.447 | 119.84 | -33.08 | 33.08 |
| CL06 | 112.37 | 141.17 | 67.65 | 0.31 | 79.60 | 83.83 | 2.19 | 0.58 | 0.33 | 0.429 | 0.310 | 89.67 | -20.20 | 20.20 |
| CL07 | 166.17 | 226.35 | 58.74 | 0.22 | 73.41 | 91.20 | 1.21 | 0.72 | 0.20 | 1.189 | 0.694 | 123.98 | -25.39 | 25.39 |
| AMA01 | 179.50 | 307.47 | 90.5 | 0.09 | 58.38 | 17.94 | 1.07 | 0.07 | 0.86 | 0.761 | 0.55 | 223.88 | 24.72 | 24.72 |
| AMA02 | 137.19 | 296.02 | 84.32 | 0.10 | 46.34 | 27.56 | 1.01 | 0.12 | 0.68 | 1.074 | 0.777 | 158.08 | 15.23 | 15.23 |
| AMA03 | 321.21 | 548.11 | 78.67 | 0.30 | 58.60 | 26.58 | 1.03 | 0.12 | 0.77 | 1.203 | 0.870 | 344.32 | 7.19 | 7.19 |
| AMA04 | 199.21 | 368.46 | 78.25 | 0.32 | 54.07 | 26.58 | 1.06 | 0.11 | 0.76 | 1.628 | 1.178 | 142.86 | -28.29 | 28.29 |
| AMA05 | 320.88 | 445.44 | 77.21 | 0.30 | 72.04 | 18.91 | 0.99 | 0.07 | 0.72 | 0.959 | 0.595 | 281.98 | -12.12 | 12.12 |
| AMA06 | 380.27 | 1535.06 | 59.22 | 0.13 | 24.77 | 16.86 | 1.46 | 0.12 | 0.20 | 1.074 | 0.777 | 374.47 | -1.53 | 1.53 |
| AMA07 | 498.91 | 2419.76 | 68.04 | 0.32 | 20.62 | 29.36 | 1.03 | 0.11 | 0.35 | 1.245 | 0.901 | 658.73 | 32.03 | 32.03 |

**Table A2.** The observed mass fractions of different aerosol compositions in Beijing, China in two previous studies, as well as the assumed fractions used in this work.

| Date | Particle size range | Sampler | $org$ fraction | $SO_4^{2-}$ fraction | $NO_3^-$ fraction | $NH_4^+$ fraction | $Cl^-$ fraction | Reference |
|---|---|---|---|---|---|---|---|---|
| Summer, 2017/2018 | PM1 | ACSM[a] | 37% | 26% | 22% | 14% | 1% | Zhou, et al., 2019 |
| Summer, 2018 | PM2.5 | ACSM | 34% | 31% | 22% | 13% | ~1% | Li, et al., 2020 |
| Summer, 2018 | 0.01~3um | PCASP | 35% | 29% | 22% | 13% | 1% | This work |

a. ACSM: Aerosol Chemical Speciation Monitor.



**Figure A1.** The profiles of observed $LWC$ and adiabatic $LWC$ for 31 liquid water cases.





**Figure A2.** The normal quantile-quantile plot for comparing the observed *w* sampled by aircraft with a standard normal distribution, for each cloud case with sufficient data. The linearity of the data points (blue dots) suggests that the observed *w* are normally distributed under a 90 % confidence level.



*Code and data availability*. The version of the QDGE scheme used to produce the results used in this paper, as well as the input data and scripts to run the model and the data to produce the key plot for the simulations, are archived on Zenodo and can be accessed at https://doi.org/10.5281/zenodo.4841035 (Wang et al., 2021).

*Author contribution*. HW processed all data, conducted all simulations and analyses, and wrote the manuscript. YP led the work, designed the experiment, and refined the manuscript. KS developed the initial model version of the QDGE scheme and provided a summary of the approach, and refined the manuscript. YY, WZ, and DZ helped with the data usage in the China campaign and refined the manuscript.

*Competing interests*. The authors declare that they have no conflict of interest.

*Acknowledgements*. The authors thank VOCALS-REx and GoAmazon2014/5 for their outstanding contributions. And the high-quality data are available on VOCALS REx official website (https://archive.eol.ucar.edu/projects/vocals/rex.html) and Earth Observing Laboratory data (https://data.eol.ucar.edu/dataset/89.132). The authors are also grateful for the supports of the Beijing Weather Modification Office and the RACA campaign. This work is supported by the National Important Project of the Ministry of Science and Technology in China (grant no. 2017YFC1501404) and the National Natural Science
Foundation of China (grant nos. 41775137 and 71690243).

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
