# Peer review of "Evaluation of a Quasi-steady state approximation of the cloud Droplet Growth Equation (QDGE) scheme for aerosol activation in global models using multiple aircraft data over both continental and marine environments"

_Geoscientific Model Development, 2021_

## Author Comment (AC1)

**Response to Referee #1 (Dr. Steven J. Ghan):**

Thanks for the careful review and instructive comments. We have revised the paper carefully based on the reviewer's comments. This is described as follows (*italic text in blue color is from the reviewer*).

*Comments:*

*This study uses in situ measurements of aerosol, updraft velocity, and droplet number to evaluate a new method for estimating cloud droplet number concentration. In addition to quantifying the mean relative error (MRE), it isolates contributions to that error from uncertainty in various inputs. This is a valuable contribution that is presented clearly, is reproducible, and of high quality.*

*However, its conclusions would be much stronger if it added, as it suggests at the end, a comparison with the performance without the quasi-steady state approximation (QSSA), i.e., using a rising parcel model with the same inputs. Without such a comparison, it is difficult to draw conclusions about the contribution of the QSSA to the MRE.*

.

**Response:**

We greatly appreciate the reviewer's comments. The reviewer affirmed the value of our work and put forward the constructive suggestion, that is, comparing the results of the QDGE scheme with the parcel model to reinforce our conclusions.

In the revised version, we examine the performance of the QDGE scheme by comparing it with parcel model results by conducting a series of experiments as described in Ghan et al., (2011). Considering different assumed aerosol types, the biases of simulated maximum supersaturations to the parcel model (i.e. the benchmark) are all below 0.18 %, showing that the QDGE scheme performs decently. Under the above premise, we carried out the closure experiments and analyzed the contributions of the QDGE to the $MRE$. The above simulations and comparison with the parcel model are included in Sect. 2.2 of the revised paper.

---

## Author Comment (AC3)

**Response to Referee #2:**

We are grateful for the careful review and instructive comments. We have revised the paper carefully based on the reviewer's comments. A point-by-point reply to the comments is described as follows (*italic text in blue color is from the reviewer*).

*Overall Appraisal:*

*This work develops a quasi-parcel model approximation to describe the activation of aerosol into cloud droplets near the cloud base of warm stratocumulus. The authors compile observations from several field campaigns around the world and use them to investigate the performance of their model. These closure studies reveal a good approximation of the parameterization to the observed cloud droplet number concentration at cloud base. This work adds to the existing pool of droplet activation parameterizations. The attempt of writing the parcel model equations on a dimensionless basis could help future development.*

*The authors place less emphasis on trying to obtain a closed analytical solution and rather use a semi-analytical integration. However, current models may be able to handle the associated computational cost. On the other hand, the exposition of the theoretical basis and rationale behind the authors approach is flimsy and, in some cases, inaccurate. These should be clarified and corrected before the work could be published.*

**Response:**

We thank the reviewer for the positive confirmation to the main goal of our work.

We agree that computing power has rapidly increased in recent years. Yet the computational cost of GCMs is still quite substantial. According to Balaji et al. (2017, doi: 10.5194/gmd-10-19-2017), the ASYPD, defined as the Actual Simulated Years Per Day for the GCMs in a 24 h period on a given platform, of models in CMIP6 ranges from 0.04 to 25.2 (median 4.9) years. The faster CPU or parallel computation helped to enhance the model efficiency, but the physical parameterizations (such as schemes to solve the cloud and radiation processes) in the large-scale grid are still the most time-consuming parts of the climate model. On the other hand, it is much more expensive for using a parcel model (currently the most accurate tool to solve the activation process) than a parameterization scheme. The computing time of a parcel model to obtain the $S_{max}$ is about several minutes, but using the QDGE scheme costs only about 0.1 seconds, and other physically-based parameterizations (such as the four state-of-the-art schemes used in Ghan et al., 2011, doi:10.1029/2011MS000074) would take even less time. For this reason, the parcel model is not practical for applying in long-term (decades or centuries) global
simulations (Ghan et al., 2011). Therefore, it is still necessary to develop
parameterized schemes to solve the aerosol activation in GCMs at present.
In addition, we elaborated our descriptions on the QDGE scheme in more detail in
Sect. 2.1. and included the comparison between results from the QDGE scheme and a
parcel model (Sect. 2.2 of the revised paper), to better explain the rationale behind the
QDGE scheme.
*General Comments:*
a)  *My main concern in this work is the lack of rationale behind the proposed*
*approach. There is very little discussion regarding the approximations taken or*
*the validity of the assumptions. Although an acceptable closure is achieved*
*against observations, this does not guarantee that the approach is theoretically*
*sound. Particularly as the evaluation of the scheme seems tightly constrained by*
*observations.*
**Response:**
Thanks for the constructive suggestion. We agree that the closure with
observational cloud cases cannot be regarded as evidence for the theoretical
rationality of the QDGE scheme. We revised Sect. 2 to improve the
methodological description of the QDGE scheme.
We included more details about the fundamental rationale of the QDGE scheme
in Sect. 2.1, explaining each approximation or assumption we have made. A
schematic diagram (Fig. 1 in the revised paper) is added to show the major steps
of the QDGE scheme. A flow chart (Fig. 2 in the revised paper) is added to
describe the iterative calculation to solve supersaturation ($S$) in each sub-level in
the QDGE scheme.
We also compare the results from the QDGE scheme and a parcel model
(following the experimental setup in Ghan et al. (2011)) to verify the performance
of the QDGE scheme (Fig. 3 in the revised manuscript). The theoretical
rationality and practical advantage (for future high-resolution GCM) of the
QDGE scheme are summarized at the end of Sect. 2.2.
b)  *The assumption of a constant saturation ratio, even over a short time step, is*
*unfounded. S changes over a very short time scale and it is not likely that it would*
*ever remain constant. Did the authors perform a timescale analysis to show*
*under what conditions their approximation would be acceptable?*
**Response:**

We are sorry for the misleading description in the previous version of the
manuscript. We assumed that $S$ was constant locally (that is, within a sub-level
with a typical height of 1~10 m) but varied with time/height throughout the host
grid of GCMs. We clarify the assumption (Sect. 2.1) and show the $S$ of each
sub-level (i.e. $S_i$ in Fig. 1b, where $i = 1, ..., N_{sub}$) in a schematic diagram (Fig.
1 in the revised paper).
In large-scale stratus clouds, the maximum supersaturation (usually less than
0.2 %) in the cloud appears about 100m above the cloud base, that is, the rate of $S$
change is $0.002 \% \, m^{-1}$ or so (Pandis et al. 1990). According to this
characteristic of aerosol activation, we assume that the supersaturation is
approximately constant in the sub-grid scale (1~10m) for the QDGE scheme. We
added the description in the revised paper (Lines 112-116, Sect. 2.1).
c)  *It is also not clear that this model can be called quasi-steady state since the*
*environment and the droplet sizes are clearly changing, and none of their*
*derivatives is negligible. What are the rigorous expressions from where the*
*parameterization is derived?*
**Response:**
The quasi-steady state refers to the following two assumptions in each sub-level:
1) the constant environmental supersaturation; 2) the conservation of total water
mass mixing ratio and liquid water static energy. In the revised paper, Eq. (1) is
the rigorous expression and Eq. (4) is the numerical expression for the particle
size growth.
The rationality of assumption 1) has been explained in our answer comment b)
above. For assumption 2), we assume that the air parcel ascends adiabatically in
each sub-level, which is the same as the assumption of the parcel model.
Correspondingly, the total water mass mixing ratio and liquid water static energy
are conservative.
d)  *The proposed model resembles a Euler integration of the regular parcel model*
*where the differential equation describing the evolution of supersaturation was*
*replaced by an iteration over an algebraic expression. The authors should*
*explain the rationale behind such approach and compare it against a more*
*rigorous model where the evolution of the supersaturation is computed explicitly*
*using a differential equation.*
**Response:**
Yes, it is a good suggestion. We added more detailed explanations on the iterative calculation in Sect. 2.1 of the revised paper. The Euler method was used to obtain

$S$ along sub-levels for approaching an $S$ profile (as shown in Fig. 1b and 1c of the revised paper). While the iteration is to calculate the $S$ value in each sub-levels.

We included more details about the fundamental rationale of the QDGE scheme in Sect. 2.1, explaining each approximation or assumption we have made.

*Specific Comments:*

*1)  Line 27. "in affecting" does not sound correct. Better say "determining"*

Have corrected (Line 29 in the revised paper, similarly hereinafter).

*2)  Lines 52-53. This is a confusing sentence. Please clarify.*

Have rewritten (Lines 53-54).

*3)  Lines 58-63. This is misleading and inaccurate. Most theoretical*

*parameterizations are approximate solutions to the parcel model equations.*

*Hence they must be evaluated against the rigorous solution first. Then, they can*

*be evaluated against observations. These are not "alternatives". Both*

*approaches aim to elucidate a different aspect of the parameterization accuracy.*

Yes, we agree that the evaluation by comparing against the rigorous solution (such as the parcel model) is necessary before the validation against observations, thus we changed "Alternatively" to "However" (Line 62).

*4)  Line 70. Is the closure experiment the same as the evaluation? Please rephrase.*

The repeated part has been removed (Line 72).

*5)  Line 75. Remove "that are"*

Have corrected (Line 78).

*6)  Line 80. Aerosol is plural already.*

Have corrected (Line 83).

*7)  Lines 82-84. This is an awkward sentence. Please rephrase.*

Have rewritten (Lines 85-86).

*8)  All equations. Please choose either supersaturation or saturation ratio, but not*

*both. Changing between s and S makes things very confusing.*

Have corrected. We use $S$ to represent supersaturation uniformly (Lines 88-89).

*9)  Line 89. Sp is the droplet equilibrium saturation ratio.*

Have corrected (Line 91).

*10) Line 92. Rephrase. "The parameters A, B and C account for ... , given by,"*

Have rewritten (Appendix A).

*11) Line 104. Please clarify what water content means in this context.*

Here the "water content" means "aerosol water contents", that is the amount of water vapor uptaken by hygroscopic growth of aerosol particles, defined as the ratio of the wet aerosol volume to the dry one. We added the explanation in

Appendix A.

*12) Line 108. Different from what? Also why would this be important near water*

*saturation, when the droplet activates?*

We now move this part to Appendix A.

$\kappa$ is a parameter introduced by Petters and Kreidenweis (2007) to represent the hygroscopicity of aerosol with a variety of chemical compounds. Whenever the chemical composition of aerosol is determined, the value of $\kappa$ can be determined.

However, Petters and Kreidenweis (2007) and Kreidenweis et al. (2008) found that the calculated aerosol water content (the ratio of the wet aerosol volume to the dry one) based on $\kappa$ biased at low relative humidity for some compounds.

Therefore, the QDGE scheme accounts for the variations in $\kappa$ with relative humidity to avoid the possible biases at low relative humidity in calculating the growth of aeroso particle.

*13) Line 112. The system is missing equations describing the evolution of the*

*saturation ratio, the temperature, and the droplet size distribution. So direct*

*numerical solution would not be only expensive but impossible.*

We added Eq. (3) in Sect. 2.1 of the revised paper to describe the variation of environmental *S*. In the QDGE scheme, we calculated the variation of supersaturation with time/height by dividing the vertical grid of the host model (large scale climate model) into sub-levels, producing a supersaturation profile in the grid. More details of the major steps of the QDGE scheme are shown in Fig. 1

of the revised paper. The supersaturation in each sub-level was iteratively calculated based on temperature and total water mass (integration over the activated particle size distribution). Fig 2 in the revised paper shows a flow chart of the iterative calculation for the supersaturation in each sub-level.

*14) Line 114. I am not sure what the "non-linear behavior of the water vapor*
*saturation ratio vertical profile" means.*

That means supersaturation $S$ is non-linear varied with height, as schematically
plotted in Fig 1c in the revised paper. We modified the sentence accordingly
(Line 106).

*15) Line 116. This is contradictory to the previous statement. If S can be assumed*
*constant, how then is it that time steps much smaller than 1 s are needed?*
*Supersaturation is relaxed quickly in cloudy parcels, so this would be wrong. The*
*authors should add more explanation and justification to their assumptions. As it*
*stands it seems very ad-hoc and possibly incorrect.*

We largely modified Sect. 2.1 and 2.2 to clarify the assumptions for the QDGE
scheme. The constant supersaturation was assumed in each sub-level (typically 1
to 10 m in height) of the host model grid. An iterative calculation was conducted
in each sub-level to obtain the supersaturation. Finally, a vertical profile of
supersaturation was produced to represent the variation of $S$ with height in the
host model grid. Figs. 1 and 2 in the revised paper show the major steps and the
iterative calculation in more detail.

*16) Line 156. What are the advantages of this calculation over writing a differential*
*equation for S?*

A key for solving the differential equation for $S$ (Eq. 3) is to determine $dq_w/dt$
by integrating wet particle size distribution calculated by Eq. (1). Whereas,
solving Eq. (1) needs the solutions of Eq. (2) and (3). Therefore, there is no
analytical solution at present for the differential equation for $S$.

Our iterative calculation is trying to use a numerical method to solve this issue
and makes the $S$ in each sub-level available. We have tested that the iterative
method can converge to the desired value quickly, so it is efficient (Fig. 2).

*17) Line 164. Where exactly can you set the entrainment rate?*

The entrainment is considered to have a direct impact on the total water mass
mixing ratio $r_t$ and the liquid water static energy $h$, as shown in Eq. (13) and (14)
in the revised paper. Both the total water mass mixing ratio and the liquid water
static energy are used to calculate the sub-level supersaturation (Fig. 2 and Eqs.
8-12 in Sect. 2.1 of the revised paper).

*18) Line 166. Couldn't find any mention of this scheme in those papers.*

Since there was no paper describing the QDGE scheme before, we could not directly mention QDGE in the Arctic research. The description "A numerically efficient solution of the condensational droplet growth equation" in Mahmood et al. (2019) stands for the QDGE scheme. But there is no description in Arora et al.

(2015). Thus, we have removed this sentence in the revised paper.

*19) Figure 2. Is the observed LWC used to drive the model?*

Yes. $LWC$ is converted to $q_w$ for calculating the initial total water mass mixing ratio $r_t$ and liquid water static energy $h$ (Fig. 2 and Eqs. 8-11 in the revised paper), which are used to calculate $S_i$ in the sub-level (Fig. 1b in the revised paper).

*20) Line 262. How does this compare against integrating over the full aerosol size*

*distribution?*

As described in Lines 318-318 we weighed the total fitted aerosol number concentration by the observed aerosol number to ensure the conservation of total number concentration (i.e., the total $N_a$ integrated over the QDGE sections in

Fig. 6c is the same as the aerosol number integrated over the observed PSD in Fig.

6a).

*21) Line 276. Internally mixed aerosol is defined as a population where all particles*

*with the same size have the same composition. Please correct.*

Have corrected (Line 328).

*22) Line 310. Please explain where this comes from. Wsub and W+ represent similar*

*things. That is, each parcel moves with a given vertical velocity. A rigorous*

*approach would integrate the parameterization over the distribution of W. In*

*absence of that, a mean (in the sense of the mean value theorem) could be used.*

*That would be either W+ or Wsub, but not both.*

As illustrated by Ghan et al. (2011) (doi:10.1029/2011MS000074), updrafts are not adequately resolved in global models, so subgrid variations in updraft velocity must be taken into account. Most climate models (e.g. Lohmann et al.,

2007; Ming et al., 2007; Gettelman et al., 2008; Wang and Penner, 2009) often represent the grid updraft velocity using the sum of the large-scale grid-mean updraft velocity ($w_+$) and the subgrid variation in updraft velocity ($w_{sub}$) within the grid cell (See Ghan et al. (2011) P16 for more details). Here we use a similar approach, $w_+$ and $w_{sub}$ are obtained from the average and the standard deviation of the probability density of function (PDF) of the sampled vertical velocity from aircraft measurement on clouds (Sect. 3.2.3), as derived in Peng et al. (2005) and Meskhidze et al. (2005). Therefore we regarded $w_+$ and $w_{sub}$ as the correspondences to the large-scale grid mean and the subgrid variation of the updraft velocity.

*23) Line 322. This sounds akward. Maybe use, "using Eq.(21) into Eq. (20) we*

*obtain"*

Have rewritten (Line 374).

*24) Line 334. Awkward sentence. Maybe just say TKE is given by…*

Have rewritten (Line 386).

*25) Line 382. Please explicitly define CDNC_M and CDNC_O*

We have explained the $CDNC_M$ in more detail and explicitly defined $CDNC_O$ in

Sect. 3.3.

*26) Line 392. Is R2 this the Pearson correlation coefficient?*

$R^2$ is the square of the Pearson correlation coefficient in our research. We modified the sentence to clarify it (Line 447).

*27) Line 418. This agreement is somehow unexpected. Given the assumptions made,*

*my suspicion is the observed LWC is used to drive the parameterization which*

*along with the total aerosol number provides a strong constraint to CDNC.*

*Please clarify whether this is the case.*

In the closure experiment, $LWC$ is used to calculate the initial $r_t$ (total water mass mixing ratio) and $h$ (liquid water static energy) by converting $LWC$ to

$q_w$ (Fig. 2 and Eqs. 8-12 in the revised paper), which is used to calculate $S$ in the sub-level (Fig. 1b). However, LWC has no direct impact on $N_{CCN}$ (Fig. 1e).

Therefore, the decent performance of the QDGE scheme in the closure experiment is not determined by using the input $LWC$ from observation.

*28) Line 450. As written, Eq. (1), i.e., the droplet growth equation, does not imply this.*

*The supersaturation balance is missing.*

This is our fault, the sentence "This is consistent with the droplet growth equation"

should be "This is consistent with the change of environmental supersaturation (Eq. (3))". We have corrected it (Line 504).

*29) Line 476. How efficient? It would be appropriate to include some timing*

*benchmarks (against rigorous solutions or other commonly used*

*parameterizations) to assess the applicability of the scheme in large scale*

*atmospheric models.*

Yes, thanks for the good suggestion. We added some descriptions about the time consumption for the QDGE scheme and the parcel model in Sect.2.2 in the revised paper (Line 210). The time of a parcel model to obtain the $S_{max}$ for a cloud case is several minutes, but it is only about 0.1 seconds for the QDGE

scheme. We also added a comparison between the results of the QDGE scheme and a parcel model for different aerosol and environmental conditions (Fig. 3 and

Sect. 2.2 in the revised paper), it confirmed the good performance and acceptable accuracy of the QDGE scheme.